# Mutations in nucleotide metabolism genes bypass proteasome defects in *png-1*/NGLY1-deficient *Caenorhabditis elegans*

**Katherine S. Yanagi**[1], **Briar Jochim**[1], **Sheikh Omar Kunjo**[1], **Peter Breen**[2], **Gary Ruvkun**[2,3]*, **Nicolas Lehrbach**[1]*

**1** Basic Sciences Division, Fred Hutchinson Cancer Center, Seattle, Washington, United States of America, **2** Department of Molecular Biology, Massachusetts General Hospital, Boston, Massachusetts, United States of America, **3** Department of Genetics, Harvard Medical School, Boston, Massachusetts, United States of America

* ruvkun@molbio.mgh.harvard.edu (GR); nlerhbach@fredhutch.org (NL)

**Data Availability Statement:** All relevant data are within the paper and its Supporting Information files.

## Abstract

The conserved SKN-1A/Nrf1 transcription factor regulates the expression of proteasome subunit genes and is essential for maintenance of adequate proteasome function in animal development, aging, and stress responses. Unusual among transcription factors, SKN-1A/Nrf1 is a glycoprotein synthesized in the endoplasmic reticulum (ER). N-glycosylated SKN-1A/Nrf1 exits the ER and is deglycosylated in the cytosol by the PNG-1/NGLY1 peptide:N-glycanase. Deglycosylation edits the protein sequence of SKN-1A/Nrf1 by converting N-glycosylated asparagine residues to aspartate, which is necessary for SKN-1A/Nrf1 transcriptional activation of proteasome subunit genes. Homozygous loss-of-function mutations in the peptide:N-glycanase (NGLY1) gene cause NGLY1 deficiency, a congenital disorder of deglycosylation. There are no effective treatments for NGLY1 deficiency. Since SKN-1A/Nrf1 is a major client of NGLY1, the resulting proteasome deficit contributes to NGLY1 disease. We sought to identify targets for mitigation of proteasome dysfunction in NGLY1 deficiency that might indicate new avenues for treatment. We isolated mutations that suppress the sensitivity to proteasome inhibitors caused by inactivation of the NGLY1 ortholog PNG-1 in *Caenorhabditis elegans*. We identified multiple suppressor mutations affecting 3 conserved genes: *rsks-1*, *tald-1*, and *ent-4*. We show that the suppressors act through a SKN-1/Nrf-independent mechanism and confer proteostasis benefits consistent with amelioration of proteasome dysfunction. *ent-4* encodes an intestinal nucleoside/nucleotide transporter, and we show that restriction of nucleotide availability is beneficial, whereas a nucleotide-rich diet exacerbates proteasome dysfunction in PNG-1/NGLY1-deficient *C. elegans*. Our findings suggest that dietary or pharmacological interventions altering nucleotide availability have the potential to mitigate proteasome insufficiency in NGLY1 deficiency and other diseases associated with proteasome dysfunction.

**Funding:** This work was supported by the Grace Science Foundation (grant to NL and GR); NIH National Institute of General Medical Sciences grant R35GM142728 to NL; NIH National Institute on Aging R01AG16636 to GR. The funders had no role in study design, data collection and analysis, decision to publish, or preparation of the manuscript.

**Competing interests:** The authors have declared that no competing interests exist.

**Abbreviations:** BTZ, bortezomib; EMS, ethyl methanesulfonate; ER, endoplasmic reticulum; ERAD, endoplasmic reticulum-associated degradation; PD, Parkinson's disease; PPP, pentose phosphate pathway; PRPS1, phosphoribosyl pyrophosphate synthetase 1; UPS, ubiquitin-proteasome system.

## Introduction

The proteasome is an elaborately regulated multi-subunit protease responsible for most targeted protein degradation in eukaryotic cells [1]. The proteasome regulates the levels of most proteins and plays a key role in cellular protein quality control through destruction of damaged and misfolded proteins. Accumulation of aberrantly folded proteins is a common feature of aging and neurodegenerative disease [2]. Deregulation of the proteasome is implicated in almost all age-associated neurodegenerative diseases and may underlie increased accumulation of misfolded proteins leading to decline in cellular and organismal function during aging [3,4]. A deeper understanding how cells maintain protein homeostasis when proteasome function is impaired may lead to new strategies for treating diseases associated with proteasome dysfunction.

Under conditions of impaired proteasome function, the SKN-1A/Nrf1 transcription factor mediates compensatory proteasome biogenesis through transcriptional activation of proteasome subunit genes [5–8]. In both *Caenorhabditis elegans* and mammalian cells, SKN-1A/Nrf1 is required for survival following exposure to proteasome inhibitors, indicating that compensatory proteasome biogenesis is necessary adaptation to pharmacological proteasome inhibition [5,8]. In *C. elegans*, the Nrf1 ortholog SKN-1A is required for the viability of animals harboring non-null proteasome subunit mutations, indicating a role for the SKN-1A/Nrf1 pathway in adaptation to proteasome dysfunction [8–10]. SKN-1A/Nrf1 is synthesized as a glycoprotein in the endoplasmic reticulum (ER). Full-length N-glycosylated SKN-1A/Nrf1 is retrotranslocated to the cytosol by the ER-associated degradation (ERAD) machinery, which also serves to target cytosolic SKN-1A/Nrf1 for rapid proteasome-dependent degradation [6,11]. Thus, SKN-1A/Nrf1 is a naturally short-half-life protein that can "sense" proteasome capacity: if proteasome function is compromised, some SKN-1A/Nrf1 escapes degradation, enters the nucleus, and drives transcriptional up-regulation of proteasome subunit genes [5,6,11,12]. Two posttranslational processing events are critical for activation SKN-1A/Nrf1 after release from the ER. These are (1) a single endoproteolytic cleavage near the N-terminus carried out by the DDI-1/DDI2 aspartic protease (cleavage of human Nrf1 occurs at W103, the site of cleavage of *C. elegans* SKN-1A has not been precisely mapped but occurs at approximately 160 amino acids from the N-terminus); and (2) deglycosylation by PNG-1/NGLY1 [8,11,13–17]. Deglycosylation of SKN-1A/Nrf1 converts specific N-glycosylated asparagine residues to aspartate, and this posttranslational modification is essential for SKN-1A/Nrf1 activity at proteasome subunit gene promoters [17,18].

Mounting evidence indicates that SKN-1A/Nrf1-dependent control of the proteasome plays an important role in normal development and physiology and can alter the consequences of proteasome dysfunction in disease. In *C. elegans*, SKN-1A is induced in response to protein folding defects in the cytosol, and this response is necessary to avert age-dependent accumulation, aggregation, and toxicity of misfolded proteins including the human amyloid beta (Aβ) peptide [9]. Further, overexpression of SKN-1A is sufficient to extend lifespan, and SKN-1A is required for the increased lifespan caused by conserved longevity-promoting interventions [9,19]. In the mouse, brain-specific inactivation of Nrf1 causes severe neurodegeneration [20,21], and Nrf1/2 activators are protective in a model of spinal and bulbar muscular atrophy [22]. Nrf1 levels are reduced in affected cells of the substantia nigra in postmortem brains of Parkinson's disease (PD) patients, consistent with failure of the Nrf1 pathway in neurodegeneration [23].

Nrf1 is also a potential target for cancer therapies, as cancers often experience chronic proteotoxic stress that is countered by the ubiquitin-proteasome system (UPS) [24]. Nrf1, NGLY1, and DDI2 are co-essential in some cancer cell lines, suggesting that Nrf1-dependent

regulation of the proteasome promotes proliferation of cancer cells [25]. Further, inhibitors of NGLY1 or DDI2 may synergize with proteasome inhibitor drugs for treatment of some cancers [16,26,27].

NGLY1 deficiency is a rare autosomal recessive disorder caused by loss-of-function mutations in the NGLY1 gene. The symptoms of NGLY1 deficiency include developmental delay, sensorimotor neuropathy, movement disorder, seizures, and alacrima [28,29]. There are currently no cures or approved treatments for NGLY1 deficiency. In ERAD, PNGase deglycosylates N-linked glycoproteins prior to their destruction [30]. However, analysis in *Drosophila*, K562 cells, and yeast suggest that NGLY1 is not required for the retrotranslocation or degradation of most ERAD substrates and that NGLY1 deficiency does not lead to activation of the ER stress responses as would be expected in the case of a severe ERAD defect [31–33]. Notably, one recent study did find evidence for glycoprotein aggregation in iPSC-derived neurons that lack NGLY1 [34]. Given the absence of a strong ERAD defect, these findings indicate that defective processing of specific glycoprotein substrates with critical physiological functions is likely to be an important driver of NGLY1 deficiency symptoms. The first such substrate to be identified was SKN-1A, the *C. elegans* ortholog of Nrf1 [8], suggesting that defective regulation of the proteasome (or other Nrf1 targets) drives NGLY1 deficiency pathology. Accordingly, inactivation of NGLY1 and Nrf1 in the mouse both cause neurodegenerative phenotypes that are accompanied by accumulation of ubiquitinated proteins [20,21,35]. Interestingly, NGLY1 inactivation in rats causes neurodegenerative phenotypes and accumulation of ubiquitin conjugates despite no apparent change in Nrf1 processing or proteasome subunit levels, suggesting that NGLY1 my impact protein homeostasis in the brain by multiple mechanisms [36]. Consistent with a critical role for Nrf1, interventions that bypass or ameliorate the effects of Nrf1 inactivation alleviate NGLY1 deficiency phenotypes in mice and worms [17,18,37]. Thus, improving proteasome function or otherwise mitigating the consequences of Nrf1 inactivation may be a means to treat NGLY1 deficiency.

Here, we took a forward genetic approach to identify mutations that act as genetic suppressors of proteasome dysfunction in *png-1/NGLY1* mutant *C. elegans*. We identified mutations, affecting 3 genes, *ent-4*, *tald-1*, and *rsks-1*, that partially suppress the hypersensitivity to lethal proteasome inhibition caused by inactivation of PNG-1/NGLY1. Although SKN-1A/Nrf1 regulation of proteasome levels is unaltered, the suppressor mutations improve proteasome function and ameliorate age-dependent tissue degeneration of PNG-1/NGLY1-deficient animals. We show that one of the suppressors, *ent-4*, which encodes a putative nucleoside/nucleotide transporter, functions at the apical membrane of intestinal cells. This indicates that reducing the uptake of dietary nucleotides can ameliorate proteasome dysfunction. Conversely, we find that a nucleotide-rich diet exacerbates proteasome dysfunction if SKN-1A/Nrf1 is inactive. These data suggest that nucleotide metabolism may modify proteasome dysfunction in NGLY1 deficiency and possibly other diseases associated with proteasome dysfunction.

## Results

### Isolation of *png-1* suppressor mutants

To study NGLY1 deficiency in *C. elegans*, we used *png-1(ok1654)* mutant animals. *ok1654* is a ~1.1 kb deletion that removes sequences encoding part of the transglutaminase domain and is a null allele [38]. We will hereafter refer to animals carrying the *png-1(ok1654)* allele as *png-1Δ* mutants. Exposure to low concentrations of the proteasome inhibitor drug bortezomib (BTZ) cause lethal growth arrest in *png-1Δ* animals but does not affect development of wild-type animals [8]. Reasoning that genetic modifiers of BTZ sensitivity could reveal targets for NGLY1 deficiency therapies, we performed a large-scale forward genetic (EMS mutagenesis) screen for

mutations that suppress the larval arrest/lethality of *png-1Δ* animals exposed to low-dose 20 ng/ml (52 nM) BTZ. This concentration of BTZ does not affect development of wild-type animals but causes highly penetrant developmental arrest and lethality of the *png-1Δ* mutants (Fig 1A). The screen yielded a total of 88 independently isolated suppressor mutant strains. Through whole genome sequencing of 60 suppressor strains, we identified multiple, independent mutations in 3 different genes: *ent-4*, *tald-1*, and *rsks-1* (Fig 1B–1D). The mutant collection includes likely null alleles of all 3 genes, including deletions and premature stop codons, as well as amino acid substitutions that affect highly conserved residues. Together, these suppressor mutations suggest that loss of function(s) normally carried out by ENT-4, TALD-1, or RSKS-1 improves the ability of *png-1Δ* animals to survive in conditions of compromised proteasome capacity (S1 Fig). These proteins carry out apparently diverse functions: ENT-4 encodes a probable nucleoside transporter related to human SLC29A1/2/3, TALD-1 is transaldolase, an enzyme in the pentose phosphate pathway, and RSKS-1 is a kinase effector of mTORC1 signaling.

After outcrossing to the parental *png-1Δ* strain, we examined the growth of the suppressor mutants on standard or BTZ-supplemented media. In each case, the mutants show delayed growth under standard culture conditions. However, the growth of each *png-1Δ*; *suppressor* double mutant is enhanced when cultured in the presence of BTZ at concentrations that drastically delay the development of *png-1Δ* single mutants (Fig 1A). We conclude that although disruption of *ent-4*, *tald-1*, or *rsks-1* reduces growth rate, it allows the development of *png-1Δ* animals to proceed under conditions of proteasome inhibition by BTZ that would otherwise cause developmental arrest.

In addition to BTZ-induced larval arrest, *png-1Δ* animals are rapidly killed by exposure to BTZ during adulthood. We therefore examined whether the suppressor mutations could improve the survival of adult *png-1Δ* animals challenged with BTZ. All 3 of the suppressor mutations significantly increase survival of *png-1Δ* mutant adults exposed to 40 ng/ml (104 nM) BTZ (Fig 1E). Thus, the suppressor mutations do not only affect development, but also alleviate the sensitivity of PNG-1/NGLY1-deficient animals to BTZ-induced proteotoxic killing as adults. This suggests that the suppressor mutations are not specific to developmental progression, but rather confer resistance to the toxic effects of proteasome inhibition more generally. Using this adult lethality assay, we confirmed that inactivation of each of the suppressor genes causes increased BTZ resistance in *png-1Δ* animals using null alleles derived independently of our EMS mutagenesis screen (Fig 1F).

In carrying out these assays, we noticed that none of the 3 suppressor gene mutations increased survival of *png-1Δ* mutant animals to wild-type levels following exposure of adults to BTZ (Fig 1E and 1F). Further, there was no improvement in survival of the double mutant animals exposed to a higher (400 ng/ml, approximately 1 μM) concentration of BTZ, even though BTZ at this concentration has no effect on survival of wild-type animals (Fig 1G) [17]. We conclude that the suppressor mutations do not completely restore normal sensitivity to BTZ in the absence of PNG-1, but instead confer a partial benefit. We tested whether combining pairs of suppressor mutations could further improve *png-1Δ* mutant resistance to killing by BTZ. Combinations of 2 distinct suppressor gene mutations improves survival of *png-1Δ* mutant animals exposed to 40 ng/ml BTZ to near wild-type levels (Fig 1H). Most strikingly, combining 2 suppressor mutations significantly increases survival of *png-1Δ* animals in the presence of 400 ng/ml BTZ (Fig 1I). The double mutants we analyzed are all combinations of null or likely null alleles. Thus, this synergistic effect suggests that the suppressors do not act together in a linear pathway that determines sensitivity to BTZ, but rather modulate resilience to this proteotoxic stress via mechanisms that are at least partially distinct. However, BTZ survival of *png-1Δ* animals harboring 2 suppressor mutations is still reduced compared to the wild type,

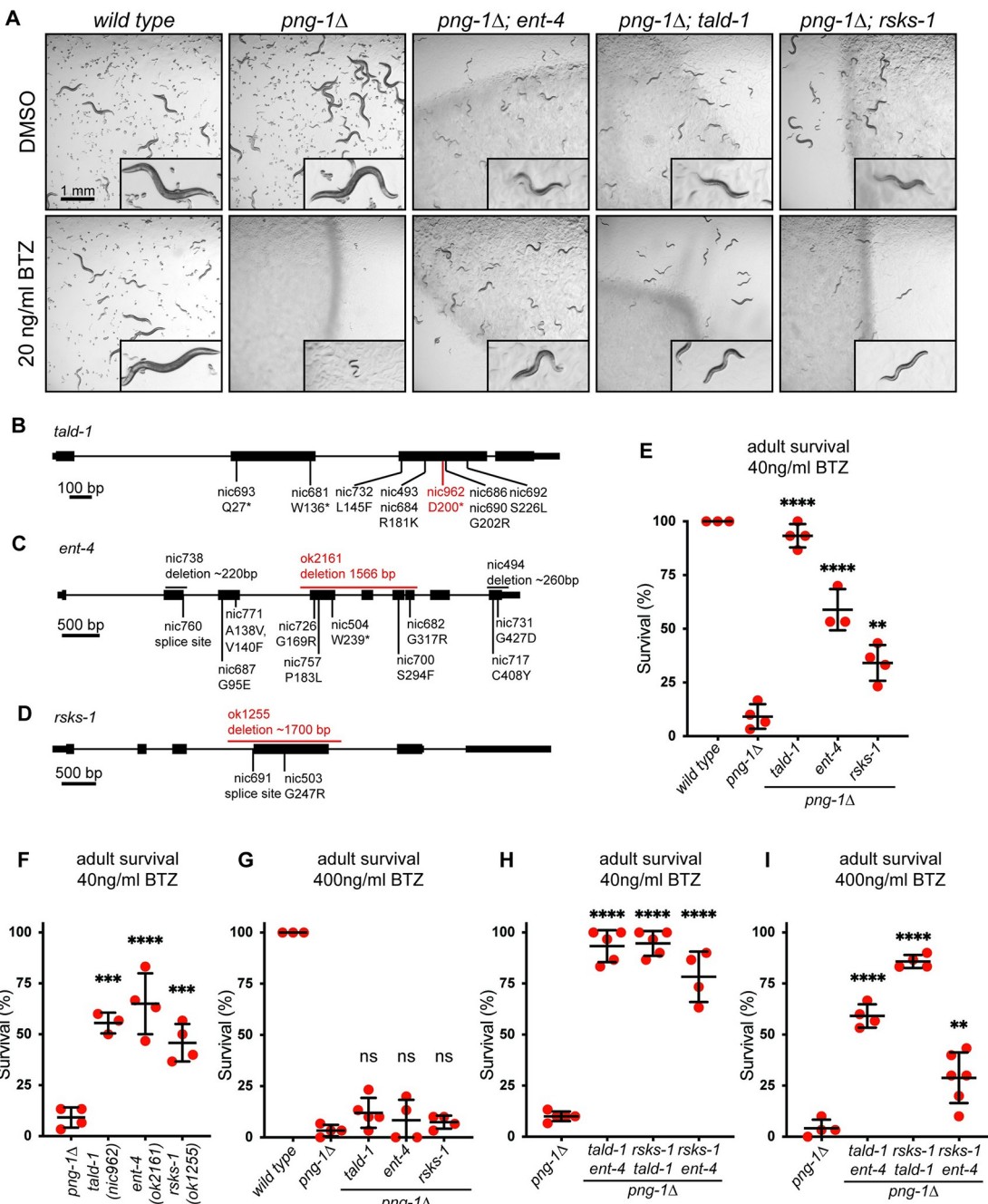

**Fig 1. Suppressors of bortezomib sensitivity in a *C. elegans* model of NGLY1 deficiency.** (**A**) Images showing growth of animals exposed to bortezomib, and 5 to 10 L4 animals of the indicated genotype were shifted to DMSO-supplemented control or bortezomib-supplemented plates. The growth of their progeny was imaged after 4 days. Each of the suppressor mutations [*ent-4* (*nic738*), *tald-1*(*nic693*), *rsks-1*(*nic503*)] restore near-normal growth of *png-1Δ* animals in the presence of bortezomib. Scale bar, 1 mm. Insets are 3× magnified. (**B–D**) *tald-1*, *ent-4*, and *rsks-1* gene models. Locations and effects of mutations isolated in a screen for suppressors of *png-1Δ* mutants' bortezomib sensitivity are shown in black text. Locations and effects of independently derived (putative null) mutations are shown in red text. The *tald-1* and *ent-4* genes are predicted to generate multiple transcript/protein isoforms, in each case only isoform A is shown. (**E**) Survival of adult animals exposed to 40 ng/ml bortezomib. The reduced survival of *png-1Δ* mutants is suppressed by *ent-4*(*nic738*), *tald-1*(*nic693*), and *rsks-1*(*nic503*). (**F**) Survival of adult animals exposed to 40 ng/ml bortezomib. The reduced survival of *png-1Δ* mutants exposed to 40 ng/ml bortezomib is suppressed by independently derived alleles of *tald-1*, *rsks-1*, and *ent-4*. (**G**) Survival of adult animals exposed to 400 ng/ml bortezomib. The reduced survival of *png-1Δ* mutants is not suppressed by *ent-4*(*ok2161*), *tald-1*(*nic693*), or *rsks-1*(*nic503*). (**H**) Survival of adult animals exposed to 40 ng/ml bortezomib. The reduced survival of *png-1Δ* mutants is suppressed by the following suppressor mutant combinations: *tald-1*

(*nic693*) *ent-4*(*ok2161*), *rsks-1*(*nic503*); *tald-1*(*nic693*), and *rsks-1*(*nic503*); *ent-4*(*nic504*). (**I**) Survival of adult animals exposed to 400 ng/ml bortezomib. The reduced survival of *png-1Δ* mutants is suppressed by the following suppressor mutant combinations: *tald-1*(*nic693*) *ent-4*(*ok2161*), *rsks-1*(*nic503*); *tald-1*(*nic693*), and *rsks-1*(*nic503*); *ent-4*(*nic504*). In panels E–I, late L4 stage animals were shifted to bortezomib-supplemented plates and checked for survival after 4 days. Results of $n$ = 3–6 replicate experiments are shown; error bars show mean ± SD. Survival of 30 animals was tested for each replicate experiment. **** $p < 0.0001$, *** $p < 0.001$, ** $p < 0.01$, ns $p > 0.05$ indicate $p$-values comparing survival rates to the *png-1Δ* control (Ordinary one-way ANOVA with Dunnett's multiple comparisons test). Numerical data for panels E-I is available in S1 Data.

indicating that the suppressors, even in pairwise combinations, are not sufficient to completely restore normal resilience to BTZ-induced lethality to *png-1Δ* mutants (Fig 1I).

## Suppressor mutants do not increase proteasome subunit levels

Loss of PNG-1 prevents the editing of N-glycosylated asparagine residues to aspartate, thus preventing the SKN-1A/Nrf1 transcription factor from transcriptionally up-regulating proteasome subunit genes [8,17]. As a result, *png-1Δ* mutants show reduced basal proteasome subunit gene expression and failure of compensatory up-regulation of the proteasome following BTZ exposure. We therefore tested if the *ent-4*, *tald-1*, and *rsks-1* suppressor mutations improve BTZ resistance by increasing the expression of proteasome subunit genes. We monitored proteasome subunit gene transcription using the *rpt-3$_p$::gfp* proteasome reporter transgene [8]. Loss of PNG-1 causes reduced basal expression of the reporter and completely abrogates up-regulation in response to 0.5 μg/ml BTZ (Fig 2A). This defect is not suppressed by loss of *ent-4*, *tald-1*, or *rsks-1* gene activities, indicating that the suppressor mutations do not cause BTZ resistance by restoring transcriptional control of the proteasome (Fig 2A). We also considered the possibility that the suppressor mutations increase proteasome levels by a posttranscriptional mechanism. As measured by western blot, the levels of the alpha subunits of the 20S proteasome are reduced in *png-1Δ* mutant animals compared to the wild type, and this defect in proteasome levels is not alleviated by any of the 3 suppressor mutations (Fig 2B). We therefore conclude that the suppressor effects on BTZ resistance are independent of regulation of proteasome levels by SKN-1A or other transcription factors.

## The suppressor mutations do not act via SKN-1/Nrf

We next sought to clarify the relationship between the suppressor mutations and *skn-1*. The fact that the suppressors do not alter the expression of the *rpt-3$_p$::gfp* proteasome subunit reporter or 20S alpha subunit levels suggests that SKN-1A/Nrf1-dependent regulation of the proteasome is not restored. However, these data do not exclude the possibility that the suppressors alter SKN-1A/Nrf1 activity to regulate other downstream targets. Indeed, SKN-1A/Nrf1 is also implicated in regulation of processes including autophagy, mitophagy, oxidative stress responses, and xenobiotic detoxification that could also contribute to BTZ resistance [37,39–41]. We therefore tested whether 2 of the suppressors (*rsks-1* and *ent-4*) also suppress the BTZ sensitivity of animals lacking SKN-1A. These mutations suppress the BTZ sensitivity of animals that lack SKN-1A identically to the *png-1Δ* mutant, confirming that the suppressor mutations do not act by altering SKN-1A activity (Fig 2C). We did not test the effect of *tald-1* in animals lacking SKN-1A because the *skn-1* and *tald-1* genes are closely linked, and our attempts to generate a double mutant by crossing were not successful.

Although only the SKN-1A isoform is capable of regulating proteasome subunit genes, the SKN-1C isoform may also contribute to proteotoxic stress resistance. We therefore examined expression of *gst-4$_p$::gfp*, a transcriptional reporter that can be activated by the SKN-1A and/or SKN-1C isoforms in response to a range of stressors [42]. Knockdown of both *rsks-1* and the pentose phosphate pathway genes *tkt-1* and *gspd-1* are known to cause activation of *gst-4$_p$::gfp*

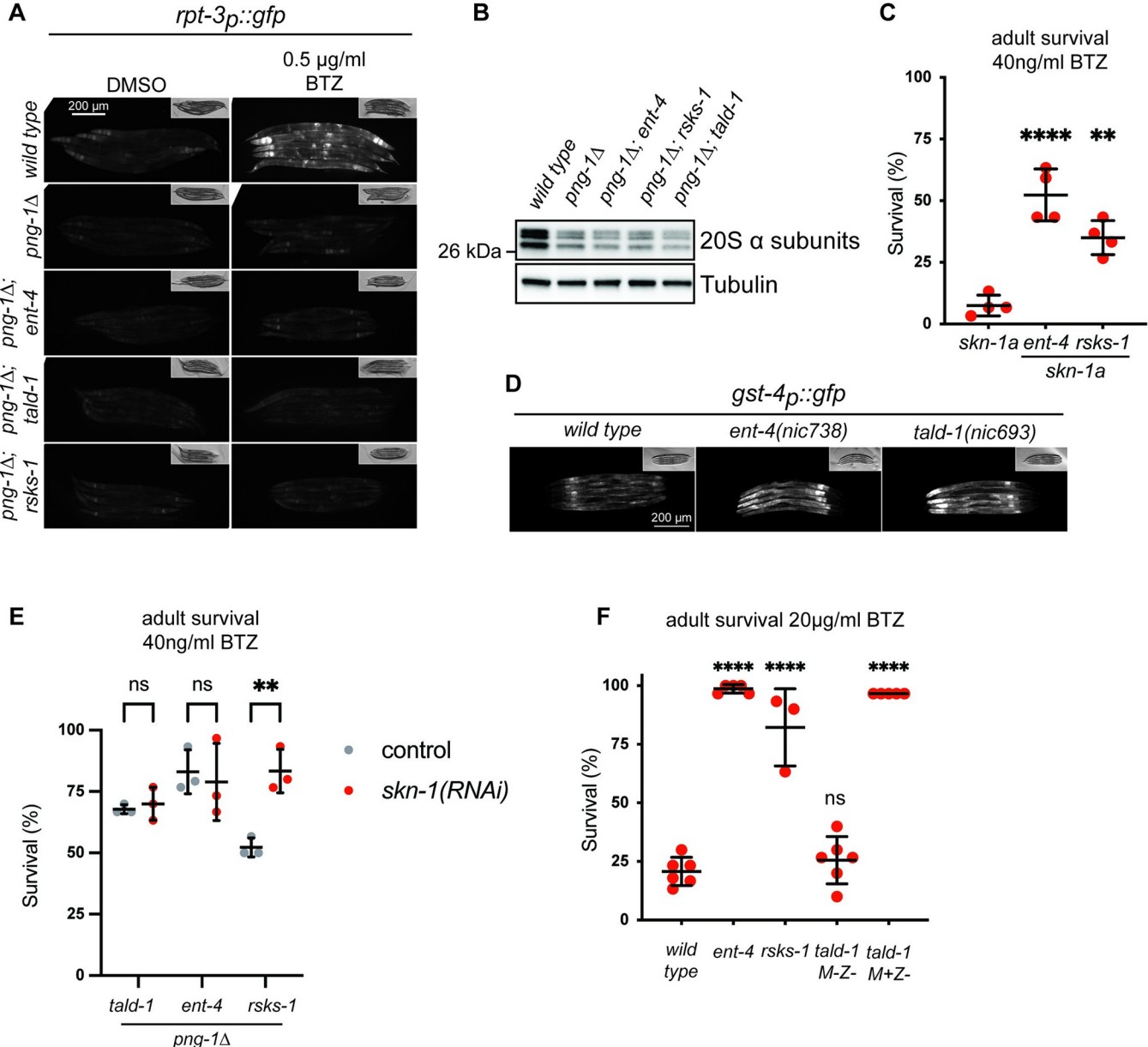

**Fig 2. The suppressors act by a SKN-1/Nrf-independent mechanism.** (**A**) Fluorescence micrographs showing expression of the *rpt-3ₚ::gfp*. Induction of *rpt-3ₚ::gfp* is defective in *png-1Δ* animals and this defect is not rescued by the suppressor mutations [*ent-4(nic682)*, *tald-1(nic693)*, or *rsks-1(nic691)*]. Scale bar, 200 um. (**B**) Western blot showing expression of 20S proteasome alpha subunits. The reduced level of proteasome alpha subunits in *png-1Δ* mutants is not altered by the suppressor mutations [*ent-4(nic504)*, *tald-1(nic693)*, or *rsks-1(nic503)*]. Tubulin is used as a loading control. (**C**) Survival of adult animals exposed to 40 ng/ml bortezomib. Late L4 stage animals were shifted to bortezomib-supplemented plates and checked for survival after 4 days. The survival of *skn-1a(mg570)* mutants is significantly increased by *ent-4(ok2161)* and *rsks-1(nic503)*. Results of *n* = 4 replicate experiments are shown; error bars show mean ± SD. Survival of 30 animals was tested for each replicate experiment. Error bar show mean ± SD. **** *p* < 0.0001, ** *p* < 0.01 (Ordinary one-way ANOVA with Dunnett's multiple comparisons test) compared to *skn-1a* control. (**D**) Fluorescence micrographs showing expression of *gst-4ₚ::gfp*. Expression of *gst-4ₚ::gfp* is increased in *ent-4(nic504)* and *tald-1(nic693)* compared to the wild type. Scale bar shows 200 μm. (**E**) Survival of adult animals exposed to 40 ng/ml bortezomib. Animals were raised on the indicated RNAi condition from hatching until late L4 stage. At the late L4 state, animals were shifted to bortezomib-supplemented plates (maintaining the indicated RNAi condition) and checked for survival after 4 days. The increased survival of *png-1Δ; tald-1(nic693)* and *png-1Δ; ent-4(ok2161)* double mutants is not affected by *skn-1(RNAi)*. In the case of *png-1Δ; rsks-1(nic503)* double mutants, *skn-1(RNAi)* causes increased survival, this is likely due to the effect of *skn-1(RNAi)* on embryo viability preventing death of *png-1Δ; rsks-1(nic503)* adults due to retention and internal hatching of eggs. Results of *n* = 3 replicate experiments are shown; error bars show mean ± SD. Survival of 30 animals was tested for each replicate experiment. ** *p* < 0.01, ns *p* > 0.05 (2-way ANOVA with Šídák's multiple comparisons test). (**F**) Survival of adult animals exposed to 20 μg/ml bortezomib. Late L4 stage animals were shifted to bortezomib-supplemented plates and checked for survival after 4 days. The suppressor mutations cause increased survival compared to the wild type [*ent-4(ok2161)*, *tald-1(nic693)*, *rsks-1(nic503)*]. Tald-1 M+Z- animals were selected from the progeny of *tald-1/tmC25* balanced heterozygotes. Results of *n* = 3–6

replicate experiments are shown; error bars show mean ± SD. Survival of 30 animals was tested for each replicate experiment. Error bars show mean ± SD. ****
$p < 0.0001$, ns $p > 0.05$ (Ordinary one-way ANOVA with Dunnett's multiple comparisons test) compared to the *wild-type* control. Numerical data for panels C, E, and F is available in S1 Data.

in a *skn-1*-dependent manner [43]. We found that *tald-1* and *ent-4* mutations both cause constitutive activation of *gst-4$_p$::gfp* (Fig 2D). We were unable to test the effect of an *rsks-1* mutation on *gst-4$_p$::gfp* expression because the reporter is linked to *rsks-1* on chromosome III, but RNAi of *rsks-1*, RNAi of other mTOR pathway genes, or treatment with rapamycin all activate this reporter and other *skn-1* target genes, strongly supporting a connection to SKN-1 regulation [43,44]. Collectively, these observations are consistent with the possibility that the suppressor mutations could impact BTZ sensitivity via activation of SKN-1C. To examine the role of SKN-1C, we measured BTZ resistance of *png-1Δ; suppressor* double mutant animals following an RNAi treatment that inactivates both SKN-1A and SKN-1C. We found that RNAi-mediated depletion of both SKN-1 isoforms did not increase the BTZ-mediated killing of *png-1Δ* animals carrying any of the suppressor mutations (Fig 2E). Unexpectedly, *skn-1(RNAi)* increased the survival of the *png-1Δ; rsks-1(nic503)* double mutant animals. This is likely because *skn-1(RNAi)* causes embryonic lethality in the progeny of the treated animals, thus preventing death of *png-1Δ; rsks-1(nic503)* adults due to retention and internal hatching of eggs. We conclude that the suppressor mutations cause BTZ resistance in *png-1Δ* animals independently of SKN-1A or SKN-1C.

## BTZ sensitivity of PNG-1-deficient animals is not dependent on AMPK signaling

Loss of RSKS-1/S6K leads to activation of AMP-activated protein kinase (AMPK) [45–47]. Intriguingly, in *Drosophila*, loss of the fly ortholog of PNG-1/NGLY1 leads to impaired AMPK signaling, and some defects of NGLY1-deficient flies can be reversed via AMPKα overexpression [48]. We therefore examined whether the BTZ sensitivity defect of PNG-1-deficient *C. elegans* is suppressed by increasing AMPK activity. We tested the role of AAK-2/AMPK in the effect of *rsks-1* and found that the *C. elegans* AMPK ortholog AAK-2 is not required for the increased BTZ resistance (S2 Fig). Thus, although AAK-2/AMPK is required for the extended lifespan of *rsks-1* animals [45], AMPK activity is not required for increased BTZ resistance in the context of a *png-1Δ* animals. In addition, we tested the effect of hyperactive AAK-2/AMPK on *png-1Δ* mutant BTZ sensitivity [49]. Overexpression of a constitutively active form of AAK-2/AMPK had no effect on survival of *png-1Δ* mutant animals exposed to BTZ (S2 Fig). Thus, increasing AMPK activity is not sufficient to increase BTZ resistance of animals lacking PNG-1. We conclude that the *rsks-1* suppressor mutations suppress the BTZ sensitivity of *png-1Δ* animals via an AMPK-independent mechanism.

## The suppressor mutations provide general resistance to BTZ

We tested whether inactivation of *rsks-1*, *tald-1*, or *ent-4* confer enhanced resistance to killing by BTZ in animals that are not deficient for PNG-1/NGLY1. We measured the survival of *rsks-1*, *tald-1*, and *ent-4* single mutant animals following exposure of adults to BTZ at 20 μg/ml, a high concentration that causes rapid killing of the wild type [17]. Unlike wild-type animals, most *rsks-1* and *ent-4* mutant animals survived at 4 days following the severe BTZ challenge (Fig 2F). Inactivation of *tald-1* also increased adult survival in this assay, in a manner dependent on the genotype of the parent. *tald-1* mutant animals only showed increased BTZ resistance if they were derived from a *tald-1* heterozygote parent (Fig 2F). This result suggests that

gene products or metabolites supplied maternally by the heterozygous parent may mask possible deleterious effects of TALD-1 loss that limit the survival of the *tald-1* mutants under proteotoxic stress. We conclude that inactivation or reduced activity of each of the suppressors confers increased resilience to the toxic effects of proteasome inhibition and this effect is not specific to sensitized genetic backgrounds in which SKN-1A/Nrf1-dependent regulation of the proteasome is defective.

## The suppressor mutations improve proteasome function

The profound effects of the suppressor mutations on BTZ resistance may reflect a reduction in the drug effectiveness through changes in drug uptake or metabolism, rather than an effect on proteostasis or proteasome function. We therefore sought to determine whether mutations in *rsks-1*, *tald-1*, and *ent-4* alleviate proteostasis defects in *png-1Δ* mutant animals that have not been exposed to BTZ. Under standard culture conditions, loss of PNG-1 causes reduced proteasome function that can be monitored by the stabilization of a fluorescent proteasome substrate Ub(G76V)::GFP [17]. Unlike wild-type animals, in which Ub(G76V)::GFP is efficiently degraded to undetectable levels, *png-1Δ* mutant animals accumulate Ub(G76V)::GFP, most prominently in intestinal cells. Strikingly, this proteasomal degradation defect is almost completely suppressed by mutation of either *tald-1*, *rsks-1*, or *ent-4* (Fig 3A and 3B).

We further explored the effect of the suppressor mutations on the UPS by examining levels of endogenous ubiquitinated proteins. Consistent with reduced basal proteasome expression levels and failure to efficiently degrade Ub(G76V)::GFP, *png-1Δ* animals show slightly increased levels of ubiquitin-conjugated proteins compared to wild-type animals (Fig 3C). Interestingly, loss of either TALD-1 or ENT-4 dramatically suppressed this increase in ubiquitin-conjugated proteins; *png-1Δ; tald-1* or *png-1Δ; ent-4* double mutants accumulate lower levels of ubiquitinated proteins than the wild type (Fig 3C). Ub(G76V)::GFP degradation is dependent on ubiquitination [50,51], so it is unlikely that *tald-1* and *ent-4* mutants are generally defective generating ubiquitin-conjugated proteins. Rather, UPS dynamics (or the activity of other critical proteostasis regulators) are altered by these mutations in a manner that broadly limits accumulation of ubiquitinated proteins. In contrast, levels of ubiquitin-conjugated proteins are slightly increased in *png-1Δ; rsks-1* double mutants (Fig 3C). It is unclear how this relates to the efficient clearance of Ub(G76V)::GFP in the mutants, but may suggest that *rsks-1* inactivation has differential effects on the synthesis, ubiquitination, and/or turnover of individual proteasome substrates.

## The suppressor mutations ameliorate age-related decline

Failure to adequately degrade damaged, misfolded, or aggregation-prone proteins is a driver of cell and tissue dysfunction in aging and late-onset neurodegenerative diseases [3,4]. In *png-1Δ* mutants, failure to appropriately regulate the proteasome is accompanied by an accelerated age-dependent decline in proteostasis and tissue homeostasis [9]. We therefore hypothesized that the suppressor mutations could ameliorate age-related defects in *png-1Δ* mutants. Age-associated loss of vulval integrity (the "Avid" phenotype), serves as a marker for age-associated decline in tissue homeostasis [52]. *png-1Δ* mutant animals show a strikingly enhanced Avid phenotype—approximately one quarter of animals undergo vulval rupture within the first week of adulthood, whereas wild-type animals only rupture at a much more advanced age [9]. The severe Avid phenotype of *png-1Δ* mutants is almost completely suppressed by loss of either ENT-4 or TALD-1, suggesting a dramatic suppression of the accelerated age-dependent decline in health and tissue homoeostasis caused by the loss of PNG-1 (Fig 3D). We were unable to test age-dependent changes in vulval integrity in the *rsks-1* mutant background,

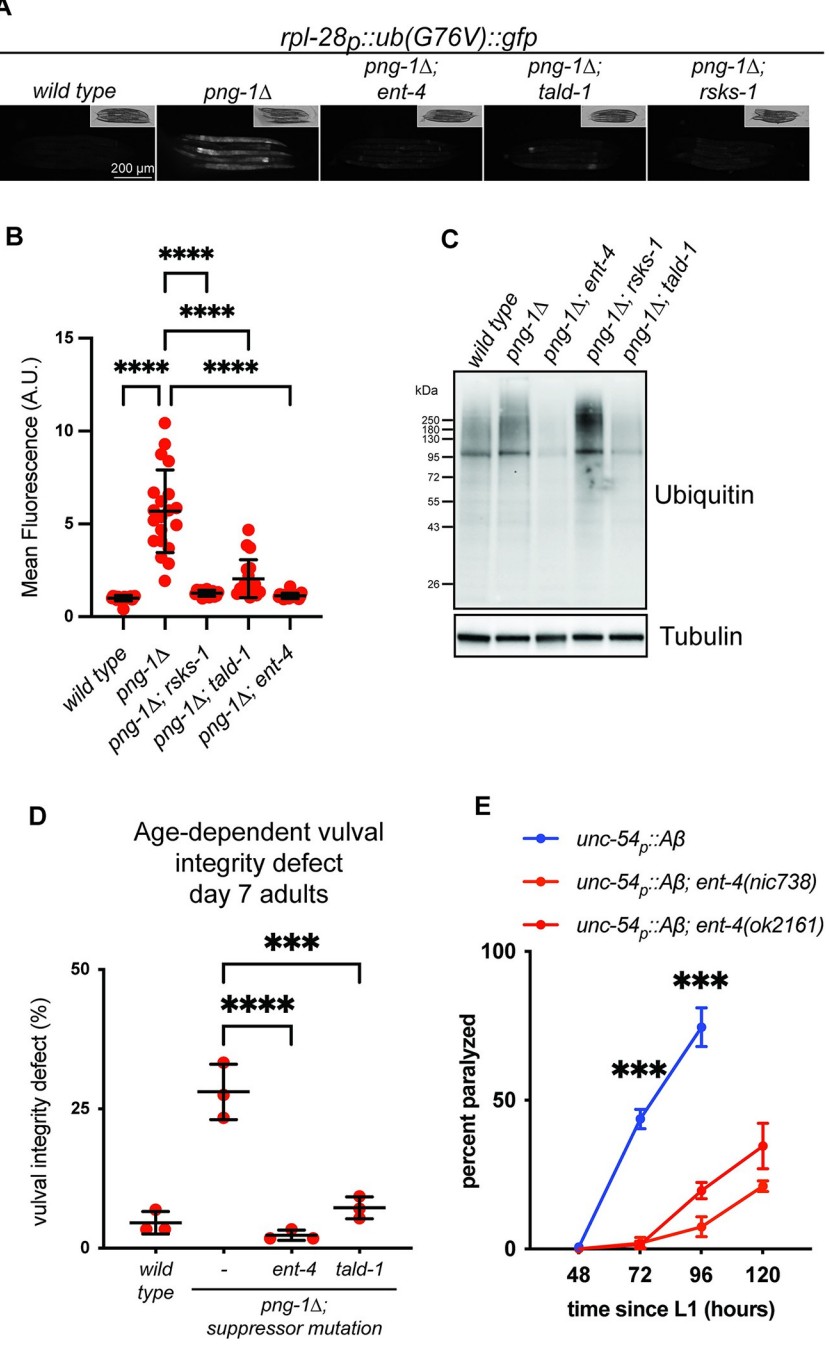

**Fig 3. The suppressors enhance proteostasis and ameliorate age-related decline.** (**A**) Fluorescence micrographs showing stabilization of Ub(G76V)::GFP *in png-1Δ* animals is reduced by the suppressor mutations [*ent-4(ok2161)*, *tald-1(nic693)*, *rsks-1(nic503)*]. Scale bar shows 200 μm. (**B**) Quantification of Ub(G76V)::GFP levels *n* = 15–20 L4 animals imaged as shown in panel A. Error bars show mean ± SD. **** $p < 0.0001$ (Ordinary one-way ANOVA with Šídák's multiple comparisons test). (**C**) Western blot showing levels of ubiquitin conjugates are mildly increased in *png-1Δ* mutants. This effect is drastically reduced by the *ent-4(nic504)* and *tald-1(nic693)* suppressor mutations, but not *rsks-1(nic503)*. Tubulin is used as a loading control. (**D**) Analysis of age-associated vulval integrity defects (Avid) in day 7 adults. The Avid phenotype of *png-1Δ* animals is suppressed by the *ent-4(nic504)* and *tald-1(nic693)* suppressor mutations. Results of *n* = 3 replicate experiments are shown. Error bars show mean ± SD. At least 50 animals were tested in each replicate experiment. **** $p < 0.0001$, *** $p < 0.001$ (Ordinary one-way ANOVA with Šídák's multiple comparisons test). (**E**) Age-dependent paralysis of wild-type and *ent-4* mutant Aβ-expressing animals. Inactivation of ent-4 delays adult-onset paralysis caused by Aβ expression in muscle. Results of *n* = 3 replicate experiments are shown. Error bars show mean ± SD. At least 100 animals were tested in each replicate experiment. Paralysis was not examined

in wild-type animals at 120 h post-L1 because a large fraction of animals had died. *** $p < 0.001$ (indicates $p$-value for the u*nc-54$_p$::Aβ* control compared to *ent-4* mutants at each time point; 2-way ANOVA with Šídák's multiple comparisons test). Numerical data for panels B, D, and E is available in S1 Data.

since many young animals lacking RSKS-1 rupture within the first day of adulthood, possibly reflecting an effect on vulval development [53].

Because the suppressor mutations increase BTZ resistance of wild-type animals, we considered the possibility that these mutations may also enhance proteostasis in aging wild-type animals. Consistently, inactivation of RSKS-1 has been shown to ameliorate developmental defects in animals lacking the key chaperone regulator HSF-1 [54] and to reduce the burden of protein aggregates in aged animals [55]. Inactivation of TALD-1 enhances proteostasis through induction of autophagy [56]. To examine the potential relevance of ENT-4 to age-related proteostasis decline, we measured adult-onset paralysis caused by expression of the Aβ peptide in *C. elegans* muscle [57]. Interestingly, loss of ENT-4 strikingly delays paralysis of Aβ-expressing animals (Fig 3E). Taken together, these data suggest that loss of RSKS-1, TALD-1, or ENT-4 has beneficial effects on proteostasis that not only ameliorate the toxicity of BTZ but also serve to enhance tissue function and resistance to protein aggregation-associated defects in aging.

## Intestinal ENT-4 modulates BTZ sensitivity

The *ent-4* gene encodes a member of the conserved SLC29 family of nucleoside/nucleobase transporters [58]. Although the role of ENT-4 has not been studied in *C. elegans*, SLC29s in humans mediate transport of a broad range of nucleosides and nucleobases or their analogs across the plasma membrane of various cell types [59]. The expression of *ent-4* mRNA is intestine specific, suggesting ENT-4 functions specifically in intestinal cells [60–62]. To identify the subcellular localization of ENT-4 in the intestine, we generated transgenic animals expressing ENT-4 fused to GFP under control of the intestine-specific *vha-6* promoter (*vha-6$_p$::gfp::ent-4*). This GFP-tagged, intestinally expressed ENT-4 specifically localizes to the apical membrane of intestinal cells (Fig 4A). This localization pattern suggests that ENT-4 may mediate uptake of nucleosides or nucleobases from the diet. When introduced into the *png-1Δ; ent-4* mutant background, the intestinal *vha-6$_p$::gfp::ent-4* transgene restores BTZ hypersensitivity (Fig 4B). Similarly, the intestinal *vha-6$_p$::gfp::ent-4* transgene restores normal BTZ sensitivity to *ent-4* single mutants (Fig 4C). We conclude that ENT-4 acts at the apical surface of intestinal cells to control BTZ sensitivity. These data are consistent with a model that reduced uptake of dietary nutrient(s) normally transported by ENT-4, possibly nucleosides or nucleobases, alters proteostasis and enhances BTZ resistance. We note that BTZ is not a nucleoside or nucleobase analog and is unlikely to be transported by ENT-4 itself.

ENT-4 could be generally required for intestinal function and may dramatically alter uptake of many nutrients aside from nucleosides/nucleobases. Indeed, the reduced growth rate of *ent-4* mutants is consistent with a severe nutritional defect. To investigate a possible relationship between nutrient uptake and proteostasis, we analyzed how other mutations known to impact nutrient uptake may affect BTZ sensitivity. *pho-1* encodes an acid phosphatase that is secreted into the intestinal lumen and is required for processing and/or uptake of unidentified nutrients [63]. *pept-1* encodes an intestine-specific oligopeptide transporter important for acquisition of dietary amino acids [64,65]. Like *ent-4* mutants, *pho*-1 and *pept*-1 mutant animals are slow growing, presumably reflecting a common consequence of impaired nutrient uptake [63,64]. Mutation of *pho-1* strongly increases BTZ resistance of wild-type worms, whereas mutation of *pept-1* has a moderate effect that did not reach statistical significance (Fig 4D). Since PEPT-1

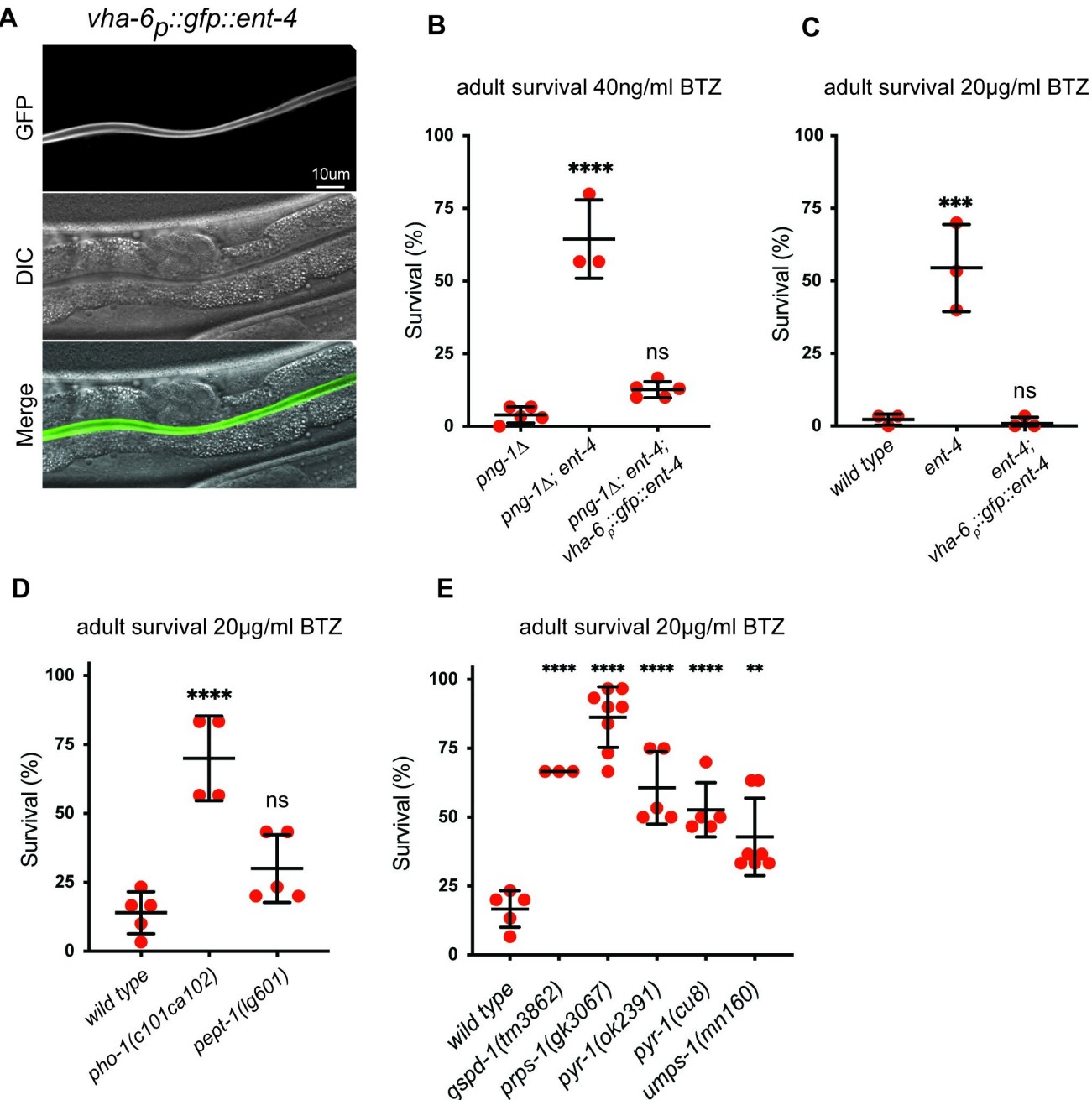

**Fig 4. Intestinal ENT-4 and nucleotide biosynthesis controls bortezomib sensitivity.** (**A**) Fluorescence micrograph showing localization of GFP-tagged ENT-4 to the apical surface of intestinal cells. Imaged animals carry a single copy Minimos insertion *nicTi354[vha-6p::gfp::ent-4::tbb-2₃′UTR]*. Scale bar shows 10 μm. (**B, C**) Survival of adult animals exposed to bortezomib, showing that intestinal expression of GFP-tagged ENT-4 (*nicTi354*) rescues the bortezomib resistance caused by *ent-4(ok2161)*. (**D, E**) Survival of adult animals exposed to 20 μg/ml bortezomib. (D) Loss of PHO-1 but not PEPT-1 increases bortezomib resistance. (E) Mutations that disrupt the PPP or nucleotide biosynthesis cause increased bortezomib resistance. For panels B–E, results of $n = 3$–5 replicate experiments are shown. Error bars show mean ± SD. The 30 animals were tested in each replicate experiment. **** $p < 0.0001$, *** $p < 0.001$, ** $p < 0.01$, ns $p > 0.05$ (Ordinary one-way ANOVA with Dunnett's multiple comparisons test) compared to the *png-1Δ* or *wild-type* control. Numerical data for panels B–E is available in S1 Data.

does not strongly alter BTZ sensitivity, increased resistance to BTZ is unlikely to be a general consequence of impaired nutrient uptake. Instead, BTZ resistance results from specific nutritional deficiencies, such as those caused by inactivating ENT-4 or PHO-1. The precise nutrients that are absorbed in a PHO-1-dependent manner are not known but might include nucleosides [63]. Thus, these data are consistent with the possibility that both PHO-1 and ENT-4 could systemically alter proteostasis via a defect in acquisition of nucleosides from the diet.

## Inhibition of nucleotide biosynthesis enhances BTZ resistance

We were intrigued by the possibility that TALD-1 may also impact proteostasis via altered nucleotide metabolism. *tald-1* encodes a pentose phosphate pathway (PPP) enzyme. The PPP is a metabolic shunt that branches from glycolysis to produce NADPH and ribose-5-phosphate (R5P). NADPH is an essential redox cofactor and R5P is required for de novo nucleotide biosynthesis [66]. First, we sought to determine whether inactivation of the PPP elevates BTZ resistance. G6PDH/GSPD-1 is essential for the first (oxidative) step of the PPP. Animals lacking GSPD-1 are highly resistant to killing when challenged with high-dose BTZ, indicating that inactivation of the PPP increases BTZ resistance (Fig 4E). Since the PPP is required for both regeneration of NADPH and providing R5P for nucleotide biosynthesis, we attempted to separate these 2 functions by examining the effect of inactivation of nucleotide biosynthesis downstream of the PPP. Phosphoribosyl pyrophosphate synthetase 1 (PRPS1) mediates an early and essential step of de novo nucleotide synthesis, converting R5P to 5-phosphoribosyl-1-pyrophosphate. Animals homozygous for a deletion allele of the *C. elegans* PRPS1 ortholog *prps-1* (systematic name R151.2) develop normally but are sterile. This is consistent with the critical role for nucleotide levels in regulating germline development [67]. Strikingly, we found that homozygous *prps-1* mutants are also highly resistant to BTZ, similarly to *gspd-1* and *tald-1* mutants (Fig 4E). We further analyzed several additional mutations that disrupt nucleotide biosynthesis at different steps, each of which enhances BTZ resistance to differing degrees (Fig 4E). Importantly, the PPP is still intact in each case (retaining the capacity for NADPH regeneration), but PPP-derived ribose can no longer be used to synthesize nucleotides. Thus, these data suggest that the BTZ resistance phenotype caused by inactivation of PPP enzymes GSPD-1 and TALD-1 are explained by their role in nucleotide biosynthesis. Taken together, these data suggest that reduced availability of nucleotides drives the BTZ resistance of 2 of the PNG-1/NGLY1 suppressors we identified: either through impaired uptake from the diet (in *ent-4* mutants) or defects in biosynthesis (in *tald-1* mutants).

## Dietary nucleotides modulate proteasome function

Our data suggest that the abundance of nucleotides may influence proteostasis in PNG-1-deficient animals. The standard laboratory diet of *C. elegans* is the *E. coli* strain OP50. In *E. coli*, cytR is a master regulator of nucleoside/nucleotide uptake and biosynthesis, serving to repress these processes under nucleotide-replete conditions [68]. The OP50 strain carries a functional copy of the cytR gene. Deletion of this gene in OP50 (ΔcytR) causes increased intracellular accumulation of nucleosides [67]. As such, the ΔcytR mutant OP50 constitutes a nucleoside-rich diet. Indeed, the additional dietary nucleosides provided to *C. elegans* that feed upon ΔcytR mutant OP50 *E. coli* are sufficient to rescue defects in development and fertility caused by mutations affecting nucleotide biosynthesis or salvage [67].

We compared proteasome-mediated degradation of Ub(G76V)::GFP in animals raised on wild-type or nucleoside-rich ΔcytR mutant *E. coli*. In wild-type *C. elegans*, altering dietary nucleoside availability has no effect on proteasome function, as indicated by the efficient

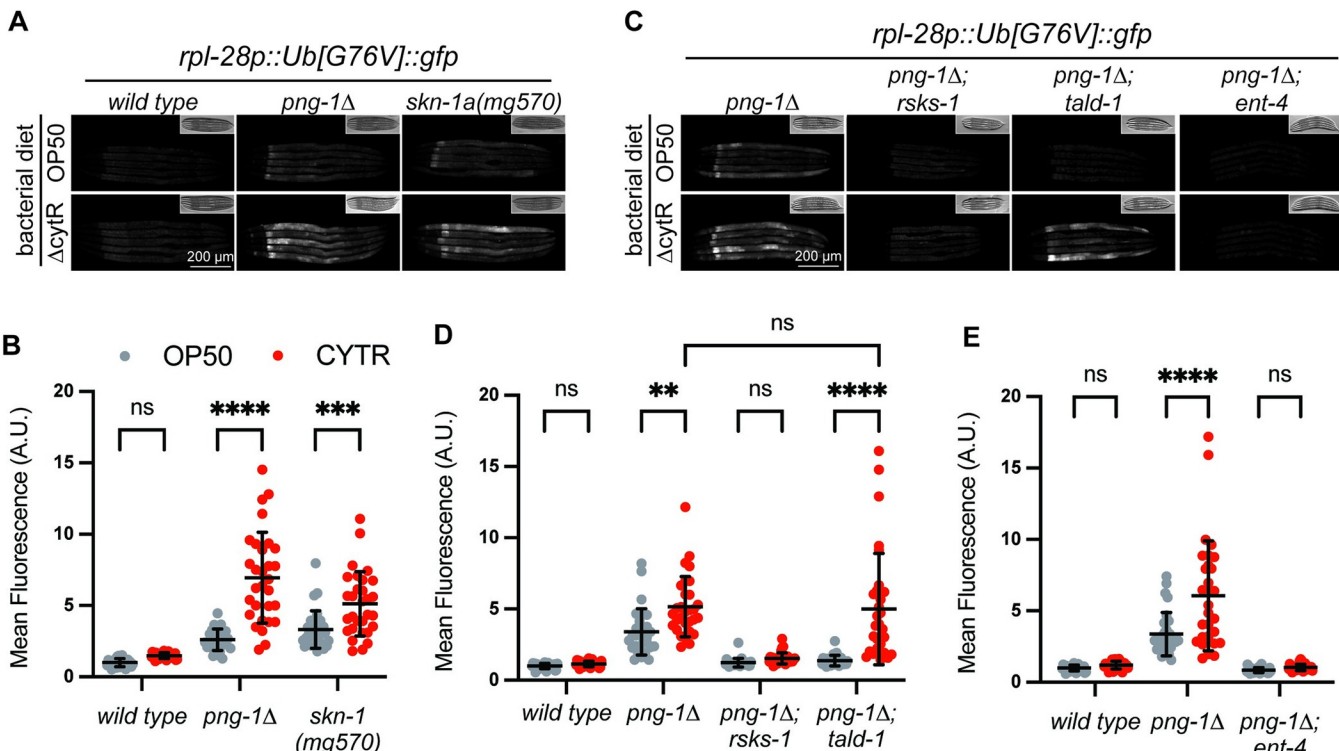

**Fig 5. Increased dietary nucleotide availability enhances proteasome dysfunction in PNG-1/NGLY1 deficient animals.** (**A, B**) Fluorescence images (panel A) and quantification (panel B) showing the effect of a nucleotide-rich diet (ΔCytR OP50 bacteria) on accumulation of Ub(G76V)::GFP. The ΔCytR OP50 diet does not alter Ub(G76V)::GFP levels in wild-type animals, but causes increased accumulation of Ub(G76V)::GFP in *skn-1a* and *png-1Δ* mutants. Scale bar shows 200 μm. (**C–E**) Fluorescence images (panel C) and quantification (panels D and E) showing the effect of a nucleotide-rich diet (ΔCytR bacteria) on accumulation of Ub(G76V)::GFP. The nucleotide-rich diet does not alter Ub(G76V)::GFP accumulation in *png-1Δ; ent-4(nic738)* or *png-1Δ; rsks-1(nic503)* animals, but increases Ub(G76V)::GFP accumulation in *png-1Δ; tald-1(nic693)* to levels similar to *png-1Δ* single mutants. Scale bar shows 200 μm. For B, D, E: *n* = 30 L4 animals per condition. Error bars show mean ± SD. **** *p* < 0.0001, *** *p* < 0.001, ** *p* < 0.01, ns *p* > 0.05 (2-way ANOVA with Tukey's multiple comparisons test). Numerical data for panels B, D, and E is available in S1 Data.

degradation of Ub(G76V)::GFP by animals fed either diet (Fig 5A and 5B). As expected, the *skn-1a* or *png-1Δ* mutant animals display a mild defect in proteasome function when raised on the standard diet of wild-type *E. coli* (Fig 5A and 5B). Strikingly, the nucleotide-rich diet led to a significant enhancement of this mild defect, causing a >2-fold increase in the accumulation of Ub(G76V)::GFP compared to animals of the same genotype fed the standard laboratory diet (Fig 5A and 5B). These data suggest that wild-type animals maintain proteostasis regardless of dietary nucleotide availability but animals lacking PNG-1 or SKN-1A are sensitive to modification of proteostasis by dietary nucleosides.

We next tested the relationship between the *png-1* suppressor mutations' effects on proteostasis and dietary nucleoside availability. As observed earlier, each of the suppressor mutations improves proteostasis in the *png-1Δ* mutant background, resulting in reduced accumulation of the Ub(G76V)::GFP proteasome substrate. However, not all the suppressor mutations maintained this improvement following feeding upon the nucleoside-rich ΔcytR *E. coli*. While inactivation of *tald-1* lowers the accumulation of Ub(G76V)::GFP in *png-1Δ* animals fed wild-type *E. coli*, it had no effect on the Ub(G76V)::GFP levels in *png-1Δ* animals fed the nucleoside-rich ΔcytR diet (Fig 5C and 5D). This is consistent with a model in which the enhanced proteostasis conferred by *tald-1* mutations derives, at least in part, from reduced availability of endogenously synthesized nucleotides/nucleosides, and this reduced nucleotide pool is reversed by increasing their availability via the diet. In contrast, the effect of *ent-4* and *rsks-1* on Ub

(G76V)::GFP levels in *png-1Δ* are diet independent (Fig 5C–5E). It is likely that *ent-4* mutant animals cannot absorb the nucleosides from ΔcytR *E. coli*, preventing dietary nucleotide availability from altering proteostasis. The fact that animals lacking RSKS-1/S6K are also insensitive to the effects of dietary nucleosides on Ub(G76V)::GFP levels suggests that signaling through RSKS-1/S6K may regulate proteostasis downstream of nucleoside/nucleotide availability.

## Discussion

Following the identification of the first NGLY1 deficiency patients, studies in cells and whole animal models have proven valuable in discovering NGLY1-dependent cellular pathways and potential means for treatment [69]. In particular, the discovery that the proteasome transcriptional regulator SKN-1A/Nrf1 is PNG-1/NGLY1-modified has prompted interest in the possibility of treatments that modulate the activity of Nrf1 or its downstream targets. Here, we have isolated mutations that suppress the sensitivity to proteasome inhibition in *C. elegans png-1/* NGLY1 mutants in the anticipation that we may identify pathways or processes that could be targets for therapeutic interventions. This work is one of several studies that have taken genetic or pharmacological approaches to identify suppressors of NGLY1 deficiency-associated phenotypes in whole animal NGLY1 mutants [18,37,48,70,71]. Interestingly, although our study is the first to point to nucleotide/nucleoside metabolism as a potential modifier of disease phenotypes, it is not the first to make a link to the composition of the diet. Developmental delay caused by inactivation of the PNG-1/NGLY1 ortholog *Pngl* in *Drosophila* can be partially suppressed by dietary supplementation of N-acetylglucosamine [32]. Also in flies, another study found that lethality caused by *Pngl* inactivation can be rescued by an isocaloric high-fat diet [72]. In mice, liver-specific inactivation of Ngly1 enhances liver dysfunction in animals provided a high-fat diet [73]. The diversity of dietary and molecular modifiers of NGLY1 deficiency phenotypes is notable and may reflect the wide range of biological processes that are impacted in this disease.

Our screening strategy exploited the sensitivity of PNG-1/NGLY1-deficient animals to lethal inhibition of the proteasome by BTZ [8]. Importantly, our results demonstrate multiple instances in which the suppressor mutations confer proteostasis benefits in animals that have not been exposed to BTZ. We therefore conclude that the primary effect of the suppressor mutations is not an alteration in BTZ drug uptake or metabolism. Instead, loss of RSKS-1, TALD-1, and ENT-4 are generally enhancers of proteostasis capacity, and consequently improve the development and survival of the *png-1Δ* mutants under BTZ challenge. Our analysis of the model Ub(G76V)::GFP substrate suggest these changes include a correction of the defect in proteasome-dependent protein degradation normally caused by PNG-1/NGLY1 inactivation, which occurs without increasing proteasome levels. The capacity to mitigate defects in protein degradation may explain BTZ resistance as well as beneficial effects on aging and proteostasis. In fact, inactivation of 2 of our hits, *tald-1* and *rsks-1*, have been linked to lifespan extension [45,56,74]. We show that inactivation of the third hit, *ent-4*, is protective in the context of the age-dependent toxicity of the Aβ peptide. Therefore, our findings may have a wider implication for modulation of proteostasis defects in aging and age-associated neurodegenerative diseases.

ENT-4, TALD-1, and RSKS-1 perform seemingly diverse functions. However, our analysis suggests a common link of each of these genes to nucleotide metabolism. We show that mutations that likely disrupt the uptake of dietary nucleotides/nucleosides or de novo nucleotide synthesis pathways cause BTZ resistance. Based on this finding, we explored the potential connection between dietary nucleotides and proteasome dysfunction in PNG-1/NGLY1-deficient animals. Strikingly, we find that a nucleotide-rich diet exacerbates proteasome dysfunction specifically in PNG-1/NGLY1-deficient animals, but not the wild type. Thus, our data suggest

that dietary nucleotides may modulate the pathology caused by loss of proteostasis in NGLY1 deficiency.

Although we have not determined the mechanism(s) that link nucleotide/nucleoside availability to proteostasis, we speculate that changes in nucleotide abundance may be sensed and coupled to regulatory programs that globally control protein synthesis and/or turnover. Our suppressor mutant collection included multiple alleles affecting *rsks-1*, the *C. elegans* ortholog of Ribosomal protein S6 Kinase (S6K), a critical effector of mTORC1 signaling [75]. mTORC1 performs a multitude of functions that govern cellular metabolism and growth, including profound effects on both protein synthesis and turnover [76]. The mTOR inhibitor rapamycin increases proteasome inhibitor resistance in mammalian cells suggesting a conserved effect of reduced S6K activity on proteasome dysfunction [77]. Interestingly, mTORC1-S6K signaling is responsive to the availability of purine nucleotides [78,79] and can also directly stimulate nucleotide biosynthesis [80,81]. Thus, S6K is both a regulator and sensor of nucleotide availability in addition to its effects on protein metabolism. The fact that *png-1Δ* animals lacking RSKS-1/S6K are insensitive to the detrimental effect of a nucleotide-rich diet on Ub(G76V):: GFP turnover suggests that *rsks-1* is required downstream or in parallel to diet-derived nucleotides to modulate proteostasis in NGLY1 deficiency (Fig 5C and 5D). However, mTORC1-S6K signaling cannot entirely explain the connection between diet-derived nucleotides and proteostasis, since loss of the putative nucleoside transporter ENT-4 still enhances BTZ resistance of *png-1Δ* animals that lack RSKS-1 (Fig 1E–1I).

A nucleotide-rich diet further disrupts proteostasis of PNG-1/NGLY1 and SKN-1A/ Nrf1-deficient animals but has little impact on the wild type, as measured by Ub(G76V)::GFP accumulation (Fig 5A). Thus, these data also imply that SKN-1A/Nrf1 promotes robust function of the UPS across variations in diet or metabolism. Indeed, Nrf1 drives increased proteasome levels in serum-stimulated mammalian cells consistent with a conserved role in coupling proteasome capacity with metabolic state [82]. Taken together with evidence that SKN-1A/ Nrf1 regulates metabolic processes including lipid metabolism, mitochondrial function, and cellular redox balance [19,73,83,84], these data suggest that PNG-1/NGLY1-dependent SKN-1A/Nrf1 signaling is a regulatory hub that coordinately regulates metabolism and protein turnover.

## Materials and methods

### *C. elegans* maintenance and genetics

Unless otherwise noted, *C. elegans* were maintained on standard media at 20˚C and fed *E. coli* OP50. A list of strains used in this study is provided in S1 Table. RNAi was performed as described [85]. *png-1*(*ok1654*), *rsks-1*(*ok1255*), *ent-4*(*ok2161*), and *prps-1*/R151.2(*gk3067*) were generated by the International *C. elegans* Gene Knockout Consortium [86]. Ethyl methanesulfonate (EMS) mutagenesis was performed by treating approximately 10,000 L4 png-1Δ mutant with 47 mM EMS for 4 h at 20˚C. F2 animals were screened for survival/growth on plates containing either 10 or 20 ng/ml BTZ. For experiments with nucleotide-rich diets, saturated OP50 or ΔCytR cultures were seeded onto plates and allowed to dry overnight at room temperature. Four L4 animals were picked onto ΔCytR or OP50 plates and animals from the subsequent generation were used for Ub(G76V)::GFP imaging.

### Identification of EMS-induced mutations by whole genome sequencing

Genomic DNA was prepared using the Gentra Purgene Tissue kit (Qiagen, #158689) and genomic DNA libraries were prepped using the NEBNext genomic DNA library construction kit (New England Biolabs, #E6040). Libraries were sequenced using an Illumina Hiseq

instrument and deep sequencing reads were analyzed using a custom Galaxy workflow adapted from Cloudmap [87]. Candidate causative alleles were identified by the isolation of multiple independent mutations affecting a given gene.

## CRISPR/Cas9

CRSIPR/Cas9 genome editing was performed by microinjection of guideRNA/Cas9 RNPs (IDT #1081058, #1072532, and custom cRNA) and ssDNA oligos (IDT) homology-directed repair templates [88,89]. The *tald-1(nic926[D200STOP])* allele was generated using a guide RNA targeting the sequence: TACACTCGAAAAGACGATCC, and the following ssDNA repair template: CGATCAGAAGGCCTACACTCGAAAAGACtgatCAAGTTGTCAGTGT-TACTCGAATCTTTA. The edit replaces the D200 codon with a premature termination codon, additionally alters the reading frame, and creates a BclI restriction site to facilitate genotyping. The edit was confirmed by Sanger sequencing.

## Transgenesis

The NEBuilder HiFi DNA assembly kit (New England Biolabs #E2621L) was used to generate pNL397 (vha-6$_p$::GFP::ent-4::tbb-2$_{3'UTR}$) by insertion of PCR products into the pNL43 [8]. The genomic DNA of *ent-4* coding sequence was amplified in 2 fragments: the first fragment spans from the start codon of exon 1 of *ent-4*, isoform B to the end of exon 3 (1,078 bp). The second fragment spans from the beginning of exon 4 to the stop codon (at the end of exon 9; 2,426 bp) of *ent-4*, isoform B. These fragments were fused in-frame and downstream of GFP coding sequence. The GFP::*ent-4* coding sequence was inserted between the *vha-6* promoter (934 bp upstream of the *vha-6* start codon) and *tbb-2* 3′ UTR (376 bp downstream of the tbb-2 stop codon). Transgenic animals harboring a single-copy insertion of *vha-6p*::GFP::*ent-4*::*tbb-2* were generated by MiniMos, using *unc-119* rescue to select transformants [90].

## Microscopy

Low magnification brightfield and GFP fluorescence images (*rpt-3$_p$*::*GFP*, *gst-4$_p$*::*GFP*, *and rpl-28$_p$*::*ub(G76V)*::*GFP*) were collected on a Leica M165FC equipped with a Leica K5 sCMOS camera and using LAS X software. High magnification DIC and GFP fluorescence images showing GFP::ENT-4 localization were collected using a Ziess AxioImage Z1 microscope equipped with an Axiocam 705 CMOS camera and using ZEN software. Worms were immobilized for imaging using sodium azide and mounted on 2% agarose pads. For all fluorescence images, images shown within the same figure panel were collected using the same exposure time and were then processed identically. To quantify Ub(G76V)::GFP fluorescence, mean pixel intensity within the anterior intestinal cells of L4 animals was measured. All Image processing and analysis was performed using Fiji software [91].

## Bortezomib sensitivity assays

Bortezomib sensitivity was assessed by the ability of animals to develop, or survive as adults, on plates supplemented with various concentrations of bortezomib (LC Laboratories, #B1408). Plates were supplemented with bortezomib by spotting a bortezomib solution on top of OP50-seeded NGM plates. The bortezomib solution was allowed to dry into the plates overnight before beginning the assay. All treatment conditions contained less than 0.1% DMSO. For developmental assays, 5 to 10 L4 animals were picked to a fresh plate supplemented with 20 ng/ml (52 nM) bortezomib. Plates were imaged to assess the growth of progeny after 4 days. For adult survival assays, 30 L4 animals were picked to fresh plates supplemented with 40 ng/

ml (104 nM), 400 ng/ml (1.04 μm), or 20 μg/ml (52 μm) bortezomib. The survival of these animals was scored after 4 days. For strains that develop on the tested concentration of bortezomib, animals were transferred to fresh bortezomib supplemented plates after 2 to 3 days to ensure that the animals being assayed for survival could be distinguished from their progeny.

## Western blot

We obtained large, synchronized populations by using bleach to isolate embryos and synchronize *C. elegans* cultures at the first larval stage by starvation overnight in M9 buffer. Approximately, 2,000 L1 animals were added to 9 cm plates seeded with OP50 and allowed to develop until the L4 stage. L4 animals we washed 3 times with M9 to remove excess bacteria and the washed worm pellet was snap frozen in liquid nitrogen. Samples were stored at −80°C until use. Worm pellets were thawed on ice and mixed with an equal volume of Bolt LDS Sample Buffer (Novex, #B0007). The mixture was heated at 95°C for 10 min and then centrifuged at max speed and 4°C for 10 min to pellet debris. Samples were run using the Invitrogen Mini Gel System on pre-cast Bolt 4% to 12% Bis-Tris Plus gels (Invitrogen, #NW04122BOX). The iBlot2 transfer device was used to transfer western blots onto nitrocellulose membranes (Invitrogen, #IB3002 or #IB23001). The antibodies used were PD41 mouse anti-ubiquitin (1:5,000, Santa Cruz, #SC-8017), mouse anti-20S proteasome (1:1,000, Sigma, #ST1049), and mouse anti-tubulin (1:10,000, Sigma, #T6074), HRP-conjugated goat anti-mouse (1:10,000, Sigma #12–349). Westerns were developed using the West Dura Extended Duration Substrate (Thermo Scientific, #34075) and imaged using a Bio-Rad ChemiDoc Imaging System.

## Amyloid beta-induced paralysis assay

For each trial, at least 100 starvation-synchronized L1 worms reared at 25°C. Worms were scored for paralysis after 48, 72, 96, and 120 h. Animals were considered paralyzed if there was no movement or in response to tapping the plate or gentle prodding.

## Age-dependent vulval integrity defect (Avid) assay

L4 animals were selected at random from mixed-stage cultures that had been maintained without starvation for at least 2 generations. To avoid progeny contamination, animals were transferred to a fresh plate on days 3 and 5 of the assay. Animals were scored for vulva rupture on days 5 and 7, and the cumulative number of animals that ruptured during the first 7 days of adulthood was recorded. Any animals that ruptured or showed an egg-laying defect in the first 3 days of adulthood were censored from the analysis. At least 50 animals were scored for each genotype in 3 replicate trials.

## Protein sequence analysis

Protein sequences were aligned using MUSCLE in SnapGene software. Membrane topology of ENT-4 was predicted and visualized using Protter [92].

## Supporting information

**S1 Fig. Suppressors of bortezomib sensitivity in PNG-1/NGLY1 deficient *C. elegans*.** (**A**) Multiple sequence alignment of *C. elegans* TALD-1 (residues 134–229) with human and *E. coli* orthologs. Missense mutations isolated in the *png-1Δ* suppressor screen are labeled. Red arrows indicate conserved residues that form the transaldolase enzyme active site. (**B**) Multiple sequence alignment of *C. elegans* RSKS-1 (residues 228–287) with human, *D. melanogaster* and *Strongylocentrotus purpuratus* orthologs. The effect of the missense mutation isolated in

the *png-1Δ* suppressor screen is labeled. The red underline marks conserved residues of the kinase active site. (**C**) Predicted membrane topology of ENT-4. Locations of missense mutations isolated in the *png-1Δ* suppressor screen are highlighted. Locations of amino acid substitutions that introduce a non-polar residue into a putative transmembrane helix are highlighted in red. Locations of other amino acid substitutions are highlighted in yellow. (**D**) Multiple sequence alignment of *C. elegans* ENT-4 with human SLC29A3 and *Drosophila* Ent2. Missense mutations isolated in the *png-1Δ* suppressor screen are labeled.
(TIF)

**S2 Fig. AAK-2/AMPK does not control bortezomib sensitivity of *png-1* mutant animals.**
(**A**) Survival of adult animals exposed to 40 ng/ml bortezomib. Late L4 stage animals were shifted to bortezomib-supplemented plates and checked for survival after 4 days. The increased survival of png-1Δ conferred by *rsks-1(nic503)* does not require AAK-2. Results of $n = 3$ replicate experiments are shown; error bars show mean ± SD. Survival of 30 animals was tested for each replicate experiment. * $p < 0.05$, ns $p > 0.05$ (ordinary one-way ANOVA with Tukey's multiple comparisons test). (**B**) Survival of adult animals exposed to 40 ng/ml bortezomib. Late L4 stage animals were shifted to bortezomib-supplemented plates and checked for survival after 4 days. The survival of png-1Δ animals is not improved by hyperactivation of AAK-2. Results of $n = 3$ replicate experiments are shown; error bars show mean ± SD. Survival of 30 animals was tested for each replicate experiment. ns $p > 0.05$ (Unpaired *t* test). Numerical data for both panels is available in S1 Data.
(TIF)

**S1 Table. List of *C. elegans* strains used in this study.**
(XLSX)

**S1 Data. Numerical data for all graphs presented in the study.**
(XLSX)

**S1 Raw Images. Original Images for western blots.**
(PDF)

## Acknowledgments

We thank WormBase for genome information and curation. Some strains were provided by the CGC, which is funded by NIH Office of Research Infrastructure Programs (P40 OD010440).

## Author Contributions

**Conceptualization:** Gary Ruvkun, Nicolas Lehrbach.

**Funding acquisition:** Gary Ruvkun, Nicolas Lehrbach.

**Investigation:** Katherine S. Yanagi, Briar Jochim, Sheikh Omar Kunjo, Peter Breen, Nicolas Lehrbach.

**Methodology:** Katherine S. Yanagi, Nicolas Lehrbach.

**Supervision:** Gary Ruvkun, Nicolas Lehrbach.

**Writing – original draft:** Nicolas Lehrbach.

**Writing – review & editing:** Katherine S. Yanagi, Gary Ruvkun, Nicolas Lehrbach.

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
