## [Editor Report · Decision Letter 0]

22 Dec 2023

Dear Nic, 

Thank you for submitting your manuscript entitled "Nucleotide metabolism mutations bypass proteasome defects in png-1/NGLY1 deficient Caenorhabditis elegans" for consideration as a Research Article by PLOS Biology.

Your manuscript has now been evaluated by the PLOS Biology editorial staff as well as by an academic editor with relevant expertise and I am writing to let you know that we would like to send your submission out for external peer review.

Once your full submission is complete, your paper will undergo a series of checks in preparation for peer review. After your manuscript has passed the checks it will be sent out for review. To provide the metadata for your submission, please Login to Editorial Manager (https://www.editorialmanager.com/pbiology) within two working days, i.e. by Dec 24 2023 11:59PM.

Kind regards,

Ines

--

Ines Alvarez-Garcia, PhD

Senior Editor

PLOS Biology

---

## [Decision Letter · Decision Letter 1]

19 Feb 2024

Dear Nic,

Thank you for your patience while your manuscript entitled "Nucleotide metabolism mutations bypass proteasome defects in png-1/NGLY1 deficient Caenorhabditis elegans." went through peer-review at PLOS Biology. Please also accept my apologies for the delay iin providing you with our decision. Your manuscript has now been evaluated by the PLOS Biology editors, an Academic Editor with relevant expertise, and by two ndependent reviewers.

As you will see, the reviewers find the conclusions novel and interesting, but they also raise several issues that should be addressed. Reviewer 1 asks for the quantification of some of the experiments, and mentions that several references are missing along with controls. Reviewer 2 would like you test additional mutants to complete some experiments and also to clarify several points.

In light of the reviews, we are pleased to offer you the opportunity to address the comments from the reviewers in a revision that we anticipate should not take you very long. We will then assess your revised manuscript and your response to the reviewers' comments with our Academic Editor aiming to avoid further rounds of peer-review, although might need to consult with the reviewers, depending on the nature of the revisions.

**IMPORTANT - SUBMITTING YOUR REVISION**

3. Resubmission Checklist

a) *PLOS Data Policy*

b) *Published Peer Review*

Sincerely,

Ines

--

Ines Alvarez-Garcia, PhD

Senior Editor

PLOS Biology

Reviewers' comments

Rev. 1: Tadashi Suzuki - note that this reviewer has signed his review

Comments on “Nucleotide metabolism mutations bypass proteasome defects in png-1/NGLY1 deficient Caenorhabditis elegans” by Yanagi et al.

This manuscript describes a suppressor screening for the hypersensitivity toward proteasome inhibitor, bortezomib. They successfully identified mutations in rsks-1, tald-1, and ent-4. They showed that the suppression is not through SKN-1, and provided convincing evidence to show that, while restriction of nucleotide availability is beneficial to mitigate proteasome insufficiency, a nucleotide-rich diet resulted in exacerbation of the phenotype.

This manuscript is, in general, very well written and easy to follow, and proposes a novel concept that modulation of nucleotide availability could be a therapeutic strategy for NGLY1 deficiency or other diseases associated with compromised proteostasis, which will surely attract the wide audience of this journal. I would thus believe that, after the following points are properly revised, this manuscript will be suitable for publication in PLOS Biology.

1. Page 5, lines 107-108; NGLY1 deficiency does not lead to signs of increased accumulation of misfolded proteins or activation of ER stress responses,

-> I am just wondering whether the authors also want to mention about the following article, which indicates that many ER stress related genes are upregulated and proteostat-positive aggregates are formed in neurons derived from NGLY1-deficient iPSCs (Manole, et al., Cell Reports 2023).

2. Page 5, lines 114-116; Indeed, the neurodegenerative consequences of NGLY1 inactivation in mouse and rat brains and those reported following brain-specific inactivation of Nrf1 are similar.

-> I am a bit puzzled by this statement as Asahina et al. rather suggested that there is no obvious changes in the level of Nrf1 and their downstream proteins (ex. proteasome subunits). Probably the authors may want to cite papers of brain-specific KO mice and specifically mention which phenotypes the authors want to refer to.

3. Page 6, lines 162-163. In each case, the mutants show delayed growth under standard culture conditions.

-> Can the authors show this data quantitatively? I also wonder how the growth are affected for each double mutant tested. Do they show inverse relationship between growth defect and resistance against BTZ?

4. Page 7, lines 196-199; However, BTZ survival of png1� animals harboring two suppressor mutations --- are not sufficient to completely restore normal resilience to BTZ-induced lethality to png-1� animals (Fig. 1e).

-> I wonder whether the triple mutant (tald-1 ent-4 rsks-1) showed even greater effect than, for example, double mutant (ex rsks-1 tald-1). LD50 value could be evaluated. I am just curious if there is any specific reason why the authors are not testing the triple mutant at all.

5. Page 9, lines 257-258; Thus, although AAK-2/AMPK is required for the extended lifespan of rsks-1 animals,

-> I am wondering whether this statement needs a reference(s). Is this a well-known fact? Sorry if I miss something.

6. Fig. 4D; are pho-1 and ent-4 really in the same pathway? If so, can we expect that they do not have an additive effect? The same discussion can be applied for gspd-1 and tald-4, just to further validate the authors’ hypothesis.

Minor points:

Line 177: a re -> are

Line 510: (Fig. 1d, e) -> Can this be Fig. 1b-e? I thought the authors are mentioning the effect of ent-4 mutation (thus comparing Fig. 1b/c with Fig. 1d/e).

Line 636: png-1� -> should be italicized (png-1�) ?

Line 708: in two fragments -> they want to put “:” here (in two fragments:) ?

Lines 757-759: It may be worthwhile to indicate the dilution of each antibody used.

Statistics in Figures: It may be better to use same symbols for statistical analysis; Can the authors use the consistent symbols (**** p<0.0001, *** p<0.001, ** p<0.01, * p<0.05, ns p>0.05) throughout the figure? For example, ** p<0.01 in Fig. 2C, E; in Fig. 3D, it says *** p<0.0001 but it should read p<0.001. Please check.

Figure 1: Why different alleles of ent-4 were used in Fig. 1b, c and d? Are they expected to give essentially same consequence? I am just curious.

Figure 3C: Can the authors indicate the size of molecular weight markers? I am also wondering whether it is worthwhile checking the level of GFP by western blotting, just to confirm the consistency between the level of fluorescence.

Rev. 2:

I am reviewing Yanagi et al. In this study they perform a mutagenesis screen to identify modifiers of NGLY1 deficiency in the worm. They identified multiple suppressor modifiers, involved in nucleotide availability, that act independent of NRF1 but still rescue the proteasome defects in NGLY1 deficiency. Overall, the study is interesting and adds new insight to the field, but the manuscript is missing a number of key data. It seemed to be a bit cherry picked in what they present. Including these missing data is important.

1. You sequenced 60 suppressor strains. Did you find other suppressors besides the 3 reported genes? If you did, a table of those results would be good.

2. Please provide a multiple species alignment for ent-4.

3. Figures S1 A-C would be nice as part of Figure 1. It’s helpful to easily see what types of mutations you are identifying.

4. Figure S1 G should also be in figure 1

5. In lines 230-233 and Figure 2C, why do you not test tald-1 as well? It’s odd and a bit glaring to leave that data out. Especially when experiments report directly before and after include tald-1. Please include this data.

6. In lines 240-241 and Figure 2D, why did you not test rsks-1? Again, it’s odd when experiments directly before and after include rsks-1. Please include this data.

7. In lines 243-248 and Figure 2E, it is worth a sentence or two of explanation of why you need a rescue of lethality with skn-1 RNAi in the rsks-1/png-1 animals. I know it is addressed in the figure legend, but an assessable description (to a non worm biologist) in the results would help.

8. Line 257-258 needs a reference in regards to the lifespan extension of AMPK in rsks-1 animals.

9. Lines 332-336 and Figure 3E, why were rsks-1 and tald-1 not tested with this specific Aβ experiment. It seems that simple lifespan extension is not the same as life span extension in the presence of Aβ. If it’s already been tested for these 2 genes, then explicitly say it here. If it has not been tested. Please provide the data.

10. Lines 341-375 and Figures 4A-C, Why not also test tald-1 and rsks-1 in these intestinal experiments. I don’t see why these other two genes wouldn’t also function through the intestine or affect nucleotide transport in the intestine.

11. Line 372-374 and Figure 4D. I would not describe the pept-1 effect as no effect. It clearly looks like there is a mild rescue, albeit not statistically significant. It reaches almost 50% with some experiments.

12. Figure 4D-E – it’s odd that the data looks very bimodal in these experiments – 25% difference in some cases. Was this done in two replicates? Two different days? Two blocks? Was this effect corrected for in the statistic?

13. Lines 492-510 – This mTOR hypothesis seems easily testable. The authors should consider some genetic experiments or rapamycin based experiments to address this. This would help add some more depth to the results.

14. There have been a number of transcriptomic studies, modifier studies, and drug screens for NGLY1. Please include some discussion as to why you are identifying these modifiers now and weren’t identified previously. There doesn’t appear to be much hint for a role of nutrients in previous studies.

---

## [Decision Letter · Decision Letter 2]

2 May 2024

Dear Nic,

Thank you for your patience while we considered your revised manuscript entitled "Nucleotide metabolism mutations bypass proteasome defects in png-1/NGLY1 deficient Caenorhabditis elegans" for publication as a Research Article at PLOS Biology. This revised version of your manuscript has been evaluated by the PLOS Biology editors, the Academic Editor and one of the original reviewers.

The review is attached below. Based on this review and the Academic Editor assessment of the revision, we are likely to accept this manuscript for publication, provided you satisfactorily address the data and other policy-related requests stated below.

In addition, we would like to suggest a small change in your title for improvement:

"Mutations in nucleotide metabolism genes bypass proteasome defects in png-1/NGLY1-deficient Caenorhabditis elegans"

We expect to receive your revised manuscript within two weeks. 

*Published Peer Review History*

*Press*

Best wishes,

Ines

--

Ines Alvarez-Garcia, PhD

Senior Editor

PLOS Biology

Fig. 1E-I; Fig. 2C, E, F; Fig. 3B, D, E; Fig. 4B-E; Fig. 5B-E and Fig. S2A, B

We require the original, uncropped and minimally adjusted images supporting all blot and gel results reported in an article's figures or Supporting Information files (Fig. 2B and Fig. 3C). We will require these files before a manuscript can be accepted so please prepare and upload them now. Please carefully read our guidelines for how to prepare and upload this data: https://journals.plos.org/plosbiology/s/figures#loc-blot-and-gel-reporting-requirements

Reviewers' comments

Rev. 2:

Thank you for addressing my concerns. Everything is satisfactory now.

---

## [Editor Report · Decision Letter 3]

21 Jun 2024

Dear Dr Lehrbach,

Thank you for the submission of your revised Research Article entitled "Mutations in nucleotide metabolism genes bypass proteasome defects in png-1/NGLY1-deficient Caenorhabditis elegans" for publication in PLOS Biology. On behalf of my colleagues and the Academic Editor, Ursula Jakob, I am delighted to let you know that we can in principle accept your manuscript for publication, provided you address any remaining formatting and reporting issues. These will be detailed in an email you should receive within 2-3 business days from our colleagues in the journal operations team; no action is required from you until then. Please note that we will not be able to formally accept your manuscript and schedule it for publication until you have completed any requested changes.

PRESS

Sincerely, 

Ines

--

Ines Alvarez-Garcia, PhD

Senior Editor

PLOS Biology
